# Evaluation of serverless computing for scalable execution of a joint variant calling workflow

Aji John[1◉]*, Kathleen Muenzen[2◉], Kristiina Ausmees[3◉]

**1** Department of Biology, University of Washington, Seattle, Washington, United States of America,
**2** Department of Biomedical Informatics and Medical Education, University of Washington, Seattle, Washington, United States of America, **3** Department of Information Technology, Uppsala University, Uppsala, Sweden

◉ These authors contributed equally to this work.
* ajijohn@uw.edu

**Data Availability Statement:** Datasets and code are available at https://github.com/SWEEP-Inc/GVCF.

**Funding:** DotMote Labs provided support in the form of contractor salaries to Kathleen Muenzen. The specific role of this author is articulated in the

## Abstract

Advances in whole-genome sequencing have greatly reduced the cost and time of obtaining raw genetic information, but the computational requirements of analysis remain a challenge. Serverless computing has emerged as an alternative to using dedicated compute resources, but its utility has not been widely evaluated for standardized genomic workflows. In this study, we define and execute a best-practice joint variant calling workflow using the SWEEP workflow management system. We present an analysis of performance and scalability, and discuss the utility of the serverless paradigm for executing workflows in the field of genomics research. The GATK best-practice short germline joint variant calling pipeline was implemented as a SWEEP workflow comprising 18 tasks. The workflow was executed on Illumina paired-end read samples from the European and African super populations of the 1000 Genomes project phase III. Cost and runtime increased linearly with increasing sample size, although runtime was driven primarily by a single task for larger problem sizes. Execution took a minimum of around 3 hours for 2 samples, up to nearly 13 hours for 62 samples, with costs ranging from $2 to $70.

## Introduction

Genetic discoveries enabled by next-generation sequencing (NGS) technologies have greatly improved the diagnosis, management and prevention of human diseases [1–3]. NGS technologies make it possible to generate whole-genome sequencing (WGS) data more quickly and cost-effectively than earlier methods like Sanger Sequencing. However, the computational methods for processing and analyzing WGS data are equally important to address [4]. Given the large size and inherent imperfections of WGS data [5], analysis of these data can be prohibitively expensive and time consuming [6].

WGS data are typically processed using computational workflows, or pipelines, which enable individual tasks to be combined into a cohesive network despite wide variations in

'author contributions' section." The funder had no role in study design, data collection and analysis, decision to publish, or preparation of the manuscript.

**Competing interests:** Kristiina Ausmees and Aji John are developers of SWEEP, the workflow tool used in the publication. DotMote Labs provided support in the form of contractor salaries to Kathleen Muenzen. This does not alter our adherence to PLOS ONE policies on sharing data and materials. There are no patents, products in development or marketed products associated with this research to declare.

execution time and package dependencies between tasks [7]. While workflows can mitigate some of the challenges presented by WGS data, they introduce other potential issues such as software incompatibilities, numeric instability, data storage and compute power limitations, and data security concerns [7, 8].

Workflow management systems (WMS) address some of the challenges associated with the execution of complex pipelines, and range from simple command line tools like SnakeMake [9] to fully-managed cloud-based enterprise solutions like Closha [10] and Terra [11]. However, the generalizability and scalability of workflows within and across platforms remains an issue [12, 13]. While tools such as Cromwell and Pachyderm [14] allow users to define portable custom workflows that can theoretically be deployed on any cloud system, they also require users to set up and maintain clusters of computational resources.

Recently, serverless computing has emerged as an alternative to using dedicated cloud resources, allowing users to execute computational units in terms of functions or containers without provisioning dedicated virtual resources. WMS that are designed for serverless execution include Hyperflow, DEWE v3 [15, 16] and the Serverless Workflow Enablement and Execution Platform (SWEEP) [17]. Some cloud providers also offer orchestration tools that can be used to create serverless workflows, e.g. AWS Step Functions and Azure Durable Functions.

In this study, we used the SWEEP WMS to define and execute a best-practice short germline joint variant calling pipeline on serverless cloud services offered by AWS and Microsoft Azure. The main motivation for selecting a serverless execution model was to obtain efficient utilization of cloud resources with relative simplicity, without need to provision and maintain resources that scale according to usage. Further, SWEEP supports multiple cloud providers, allowing for a high degree of control over workflow execution while avoiding the lock-in that can result from employing vendor-specific solutions. SWEEP workflows are defined by means of a data serialization language, following a similar structure to others such as Common Workflow Language (CWL) and Workflow Description Language (WDL) that are familiar to many users in the field of bioinformatics. A major difference, however, is that the SWEEP workflow definition framework allows for the use of more complex control constructs, such as multi-level scatter and gather behaviour.

SWEEP has previously been used to successfully perform single-sample variant calling in a cohort of 128 human WGS samples [17]. In this study, we implemented another common, but computationally more demanding, variant calling pipeline as a SWEEP workflow. While other studies have demonstrated the utility of combining serverless computing and bioinformatics [18, 19], we aimed to evaluate the execution of more complex serverless workflows, and to identify potential shortcomings of this approach.

The workflow implemented in this study was based on the Broad Institute's Genome Analysis ToolKit (GATK) [20] joint variant calling pipeline for short germline variants. Single-sample and multi-sample variant calling are two methods commonly used to detect variation among genetic sequences. While multi-sample variant calling (also known as joint variant calling) improves the sensitivity of variant identification in larger sample sizes, it is more computationally expensive and less scalable than single-sample calling [21]. The GATK best-practice workflow mitigates some of the scaling issues of traditional joint variant calling workflows by allowing for a combination of single-sample variant calling and joint genotyping using a genomic VCF (GVCF) file intermediate [22]. This workflow is suitable for assessing the strengths and limitations of serverless computing in genomic analyses, given that it employs both individual and cohort-level methods, and is widely used in the genomic community. Although the total runtime of this enhanced workflow is only expected to increase linearly with increasing sample size, users may still run into resource limitations with very large sample sizes [23].

In the following sections, we describe the adaptations made to define the GATK variant calling pipeline as a SWEEP workflow, and discuss design considerations and limitations encountered. Finally, we present results from executing the workflow using up to 62 samples from the 1000 Genomes project phase III data set [24], and provide an evaluation of the scaling performance.

## Materials and methods

### Definition of workflows

SWEEP workflows are represented as Directed Acyclic Graphs (DAGs), where the nodes correspond to tasks, and the arrows indicate order of execution. Tasks make up the executable units of the workflow, and can be constructed from functions or containers. Function-based tasks are defined by code that can be run on a cloud provider (CP) in a serverless manner using the Function-as-a-Service (FaaS) execution model. Container-based tasks comprise Docker images that wrap dependencies in addition to code and scripts, and are executed using the Container-as-a-Service (CaaS) model.

For both types of tasks, a task definition containing information about the executable unit is registered to SWEEP. A task is considered registered when the task is deployed via SWEEP to a particular CP. Aside from the runnable code, the task definition may also include settings for execution control such as allocation of compute units and memory. SWEEP deploys the tasks to various CP(s) beforehand, and has the knowledge of the resource namespace for invocation. SWEEP has adapters built-in to invoke FaaS or CaaS artifacts across different CP(s).

A workflow is subsequently defined by tying together individual tasks into a workflow graph. SWEEP workflow definitions are JSON files that specify the order of task execution and contain additional orchestration information. On a task level, the latter may include conditional statements, delays, as well as error and retry behaviour. Dynamic workflow unrolling can be defined using the "scatter" construct, and flow of information through the workflow graph can be controlled by means of specifying task input and output. For container tasks, environment variables and the command to run on execution may also be specified. Global workflow properties such as constraints on runtime task concurrency can also be set, as well as specifications of the CP(s) that should be used.

As serverless execution is performed in a stateless manner, there is no persistence of data between tasks on the compute infrastructure itself. External data storage services are thus required to persist data throughout the execution of the workflow. In SWEEP, there is no built-in infrastructure for data persistence; it is up to the designer of the tasks to specify an external data storage location.

### Case study: Joint variant calling of 1000 Genomes data

The GATK best-practice joint variant calling pipeline was implemented as a SWEEP workflow comprising 18 tasks. Options for resource settings such as number of CPUs, memory and maximum runtime are dependent on the FaaS and CaaS services offered by particular CPs, with CaaS generally allowing larger resource allocations. For this reason, the more computationally intensive parts of the variant calling workflow were defined using container-based tasks, and function-based tasks were used for lighter compute loads with shorter runtimes.

The majority of tasks in the workflow are container-based tasks built from custom Docker images that were adapted to the serverless workflow paradigm, as opposed to the standard Docker Hub base images that are typically used for genomics analyses. The required package dependencies for a given container task are bundled into custom Docker images that can be deployed on virtually any computing platform. Each image is based on a Linux OS, with

installations of BWA, Picard, Samtools and GATK added as needed. The remainder of the tasks are lightweight functions that are used to implement handling of meta-information used in workflow control structures, such as scatter and gather behaviour.

Fig 1 shows the workflow DAG, with function-based tasks surrounded by dashed lines and container-based tasks by solid lines. In Task 1, the per-sample metadata required to orchestrate subsequent tasks is prepared and propagated onwards. Tasks 2–6 are then run in parallel, generating index files from the reference sequence and known SNP/indel files. The remainder of the workflow tasks rely on the successful completion of each predecessor and are run sequentially. Tasks 7–13 are scattered by sample, and collectively produce QC'd GVCF files from the paired-end read files for each sample. Task 14 takes inventory of all GVCF files that have been successfully produced by Task 12 and defines the separate task variables for Task 15, which is scattered by chromosome. Joint variant calling is performed for chromosomes 1–22 in separate container tasks, and the VCF outputs of each task are stitched together by the Picard GatherVCFs function in Task 17. The final output of the workflow is a single joint VCF file that contains SNP and indel information for each sample included in the workflow. The additional information at https://github.com/SWEEP-Inc/GVCF also contains an image of the complete unrolled workflow DAG for a problem size of 14 samples.

The variant calling workflow was executed using the Amazon AWS and Microsoft Azure CPs. All container tasks were executed on Azure Container Instances (ACI), as they were provisioned with 50GB of disk space compared to 10 GB for AWS Fargate at the time of the study. All container-based tasks were configured to use 4 CPU cores and 16 GB memory. Function-based tasks were executed on AWS Lambda and were configured with 836 MB of memory. Amazon S3 was used as an external data storage service.

## Detailed task descriptions

Given an input CSV file containing Amazon S3 URLs linking to the sample data, Task 1 generates the required metadata for scattering subsequent tasks over the samples, and returns the sample-specific information for each parallel task in terms of environment variables. Tasks 2–6 are then used to generate dictionary and index files for the input reference sequence files. Task 2 uses the BWA index function to create an index (.idx) file from the reference sequence FASTA file. The output index file is a required input for the downstream alignment step. Task 3 creates a sequence dictionary (.dict) file from the reference FASTA file using the Picard CreateSequenceDictionary function. Task 4 creates another index (.fai) file from the reference FASTA file using the Samtools faidx function, since several downstream tasks require this index format instead of the .idx format. Tasks 5–6 create index (.idx) files for known insertions/deletions and SNPs in the reference sequence using the GATK IndexFeatureFile function. Tasks 2–6 are run in parallel as they are independent of one another.

Once all index and dictionary files have been successfully generated, Tasks 7–13 are run sequentially for each sample. For each of these tasks, intermediate outputs are stored in S3. Task 7 is scattered by sample, and uses the BWA mem function to align Illumina paired-end reads for a single sample to the reference sequence, producing a BAM file intermediate. This BAM file is further processed by Task 8, which uses the Picard AddOrReplaceReadGroups function to sort the reads. Task 9 then uses the Picard MarkDuplicates function to identify DNA fragments that are duplicates of one another. The output BAM file is recalibrated in Task 10, which uses the GATK BaseRecalibrator function to identify systematic errors in base call quality scores. The output recalibration table is used in Task 11 to create a recalibrated BAM file using the GATK ApplyBQSR function. Finally, Task 12 uses the GATK HaplotypeCaller function to call germline SNPs and indels in a single BAM file. In this workflow, the tool is run

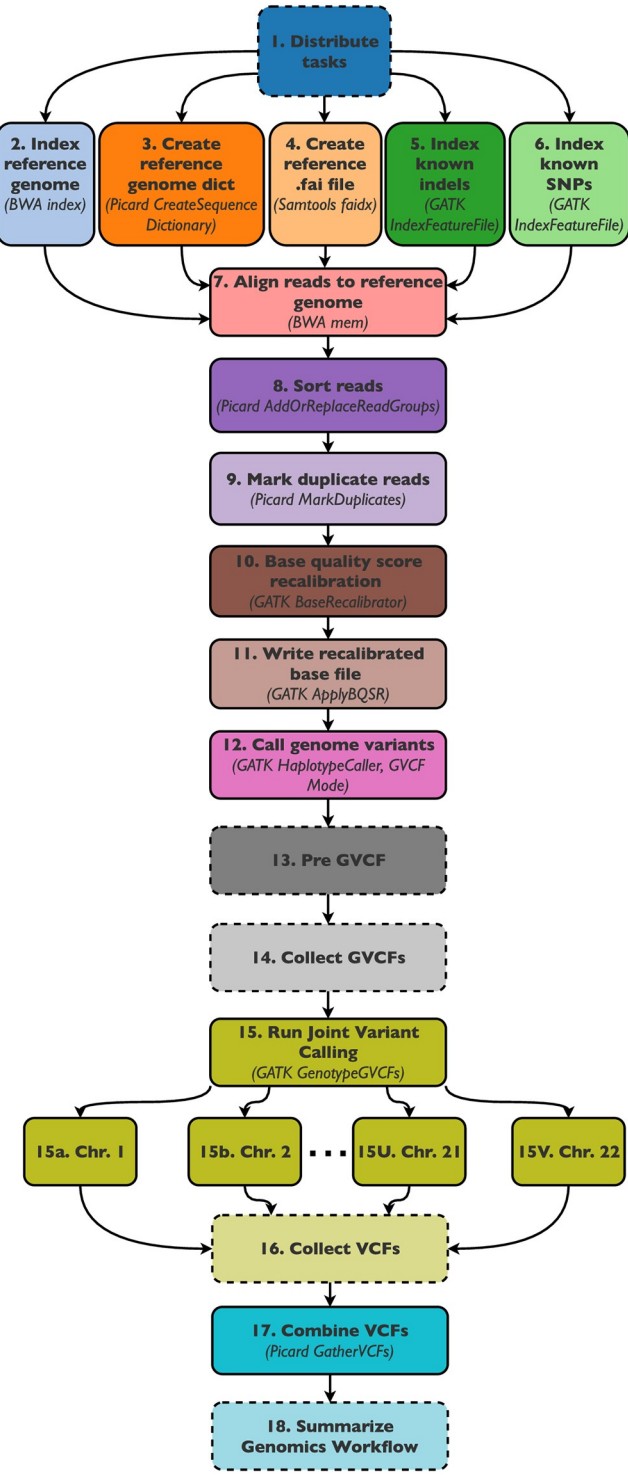

**Fig 1. Joint variant calling workflow tasks.** Diagram of the joint variant calling workflow as implemented in SWEEP. Dashed and solid edges represent function-based and container-based tasks, respectively.

in GVCF mode to produce an intermediate genomic VCF file that can later be used for joint genotyping of multiple samples.

After the per-sample variant calling has completed for each input sample, Tasks 14–17 are run in sequence to generate a single joint VCF file. Task 14 is an intermediate function-based task that collects environment variable information for all tasks that have been running in parallel up until this point in the workflow. Task 14 uses the output of Task 13 to search the S3 bucket for successfully generated GVCF files from Task 12, create a list file and map file containing information on each GVCF file, and then define the environment variables needed to parallelize task 15.

Task 15 is scattered by chromosome, and genotypes are calculated on a per-chromosome basis across all input samples using the GATK GenotypeGVCFs function. The output of each parallelized task is a single-chromosome joint VCF file. Task 16 comprises a function that generates a list file containing the S3 paths of all per-chromosome VCF files generated by Task 15. Task 17 then uses the Picard GatherVCFs function to knit together all per-chromosome VCF files into a single joint VCF file. Finally, Task 18 generates a message of success or failure upon workflow completion.

## Results

The workflow was executed on Illumina paired-end read data of samples from the European and African superpopulations from the 1000 Genomes project phase III data set. Initially, 62 samples were selected at random, and used to define overlapping subsets consisting of 2, 8, 14, 20, 26, 32, 38, 44, 50, 56 and 62 samples. The workflow was executed 5 times for each problem size, and we report the average over these runs unless otherwise specified. See the workflow artifacts at https://github.com/SWEEP-Inc/GVCF for more detailed information about the sample data.

In the case of problem sizes greater than 62, several of the tasks failed when executed on Azure. The SWEEP system can handle task failures by retrying, but we did not exercise this option in order to maintain comparable metrics across runs. According to the Microsoft Azure support, the issues experienced were related to dynamic resource provisioning due to the high disk usage, with resource instability and failures occurring even though total disk usage stayed within the specified limits of the service. For this reason, problem sizes greater than 62 are not included in the results.

The per-task distribution of runtimes in Fig 2 reflects variation within and between problem sizes for several tasks. Overall runtime may be impacted by individual and chromosomal effects on computationally intensive tasks, as well as variability due to the service level provided by the CP. However, as there is only one instance of Task 2 per run regardless of problem size, the lower variation observed suggests a relatively consistent service level in spite of relatively high runtimes. For Tasks 7 and 12, of which there is one instance per sample per run, the relatively high variability of runtime is likely driven by inherent differences in read depth and quality of the samples.

Task 15 is scattered over autosomes and combines data across all samples, and therefore has an expected linear increase in runtime as the problem size increases. This trend is visible in Fig 3, and is a plausible driver for the high variability seen in overall task runtimes in Fig 2. This also offers an explanation for the observed linear increase in overall workflow runtime seen in Fig 4. Since the computational requirements of each joint variant calling task inherently grow with problem size, this workflow does not display the ideal serverless behaviour of linear cost and constant runtime with increasing problem size. We do note, however, that overall workflow runtime could be reduced with additional parallellization by scattering Task 15 over smaller chromosomal segments.

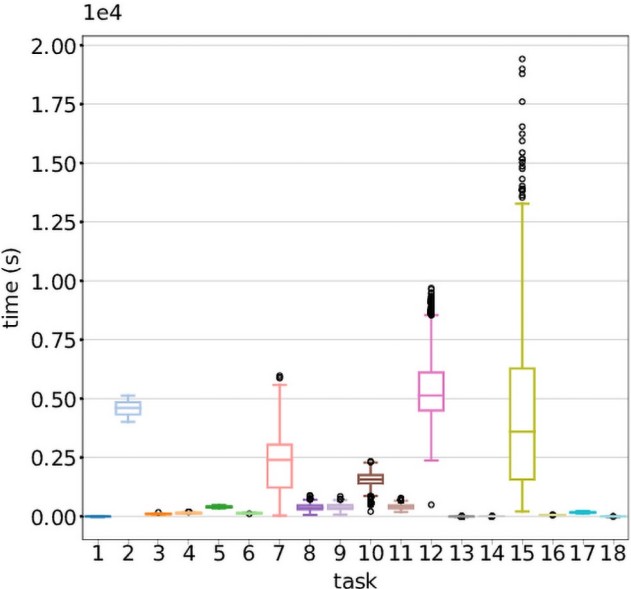

**Fig 2. Runtime of workflow tasks across all problem sizes.** Distribution of the runtimes of the 18 task types over all problem sizes and runs. Tasks 2, 7, 12 and 15 had the highest runtimes overall, as well as the largest degree of variability. Fig 4 shows the median runtime of the tasks for each problem size, averaged over the 5 runs, stacked and color-coded by task type. Again, it is visible that tasks 2, 7, 12 and 15 contribute substantially to overall workflow runtime. The median task runtime remains constant for all tasks except 15, where it grows with problem size.

## Discussion

In this study, we developed a novel serverless workflow for joint variant calling by adapting a gold-standard GATK pipeline to the SWEEP platform. Workflow execution spanned two different CPs, highlighting the possibilities of harnessing unique features across providers.

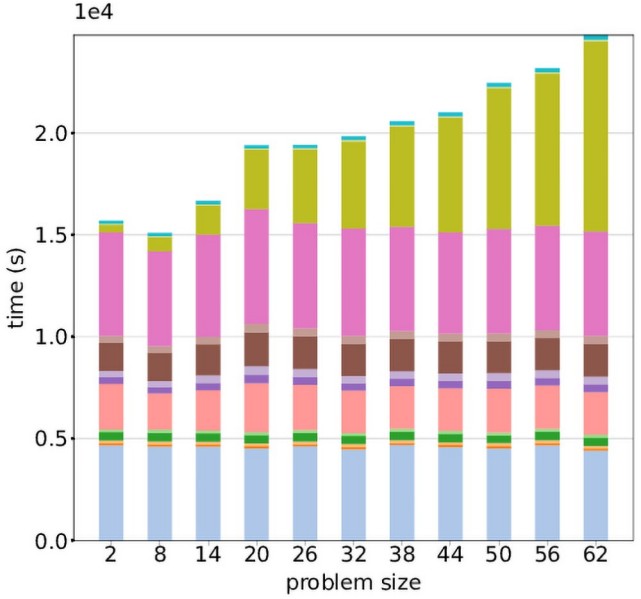

**Fig 3. Per-task runtimes across problem sizes.** Median task runtime of the different task types of the joint variant calling workflow, averaged over five runs and stacked per problem size. Task types are represented by the same colors as in Fig 1.

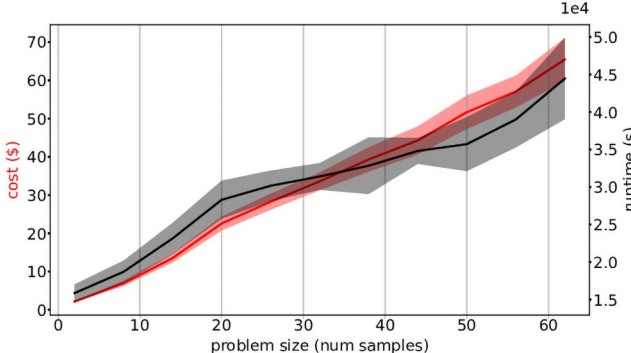

**Fig 4. Mean runtime and cost across problem sizes.** Average runtime and cost of the workflow for different problem sizes, with shaded regions indicating standard deviation. Both metrics increased linearly with problem size, and show higher variability for increasing problem sizes. Execution took a minimum of around 3 hours for the smallest problem size of 2 samples, up to nearly 13 hours for 62 individuals, with costs ranging from $2 to $70. Task types are represented by the same colors as in Fig 1.

In the serverless paradigm, the CP manages resource provisioning, and the potential parallel capacity is theoretically unlimited. In practice, the execution of the workflow experienced resource instability with increasing problem size. For the purposes of this study, this instability limited the maximum problem size to 62, although we note that larger problem sizes were possible to run with the current setup when allowing for retry of failed tasks.

Further, we noted that failures due to instability of the CP were reduced when the same compute and memory resources were allocated to all container-based tasks across runs. Although this approach resulted in marginally higher costs compared to adapting the resource allocation to problem size, we chose to maintain homogeneous allocation to obtain comparable runtime metrics for the purposes of this study. In practical applications, compute resources could feasibly be re-allocated disproportionately to tasks once the workflow development and testing phases are complete.

These issues highlight a main concern regarding the use of serverless resources; the stability and response of the computational environment are subject to variability due to the service level of the CP. Limits on number of concurrent tasks, whether pre-specified or found out empirically, invocation overhead, discussed in e.g. [25], and API throttling due to large numbers of requests are examples of factors that can affect the efficiency and feasibility of serverless execution of workflows.

There are additional limitations to executing data-intensive workflows on serverless resources. For example, AWS Fargate limited container disk space to 10 GB at the time of the experiment, which rendered some of the tasks impossible to execute. This limitation was possible to circumvent by execution of containers on ACI, which provided 50 GB of disk space. This required defining the routing of container-based tasks to ACI in the workflow definition, rather than using system-level routing in which the optimum destination of a task is chosen by SWEEP. This example illustrates how multi-cloud architectures can facilitate the adaptation of a wider range of problems to the serverless format. It also suggests the possibility for workflow performance optimization in terms of cost and runtime by designing scheduling policies that are aware of CP-specific properties, although such considerations were not taken into account in the current study.

We further note that AWS Fargate has since added support for mounting persistent volumes to increase disk space, which could help overcome the issues discussed above. We predict that the competitive nature of commercial cloud providers will lead to an evolving host of

serverless offerings, and render the serverless paradigm increasingly suitable for running compute-and data-intensive workflows.

Another important consideration in the design of serverless workflows is the statelessness of tasks and the resulting need for external persistent storage. For example, the 1000 Genomes data resides on AWS S3 as part of an open-data initiative, and the transfer of container execution to ACI may therefore not have been ideal as it increases the distance between the data and compute sources, potentially causing higher latency and costs due to data transfer across different CPs. We opted to continue using S3 to persist data between tasks, but the workflow could have been further optimized by adaptive selection of storage based on proximity to the computational resources used. In this particular scenario, this could have been achieved by mounting Azure file share volumes to persist data between the container instances on ACI, and limiting the use of S3 for cross-cloud communication between tasks. Similar support could also be done for the functions by having a file system mounted on load.

Traditionally, joint variant calling workflows are run on HPC or dedicated cloud clusters on commercial CPs [26]. Although such solutions have proven efficient in terms of runtime [27], the reproducibility of genomic workflows remains a challenge with many aspects not fully addressed [28]. As computational pipelines and execution environments grow more complex and heterogeneous, the mere release of code and data is often not enough to ensure reproducibility [29, 30]. In [30], the benefits of tools such as CWL and Docker for virtualization and encapsulation of critical workflow provenance information is discussed, but they also note that such approaches have a steep learning curve in terms of both language principles and system configuration.

The serverless paradigm thus promotes workflow reusability by simplifying the process of executing workflows consisting of virtualized components such as containers and functions, and relieving the user from the responsibility of ensuring the availability of sufficient compute and storage resources. Further, the declarative workflow definition format of SWEEP, in conjunction with the code and metadata used to execute the tasks, encapsulates all information required to reproduce the workflow, either on the SWEEP platform or another WMS.

Finally, the stateless nature of serverless workflows can enhance portability by encapsulating execution information such as use of external storage in the workflow code itself, yet could potentially increase the user's dependency on third-party resources and risk 'workflow decay' due to external resources changes [31]. As with all workflows, it remains important to document software versions, parameter settings and any assumptions made in the workflow execution to ensure reproducibility.

At the time of writing, the authors were not aware of any other WMS that supported cross-cloud serverless execution of functions as well as containers. However, a relevant comparison, which we leave to future work, would be to compare SWEEP to CP-specific orchestration solutions.

## Conclusion

An explosion of cloud-based storage and analysis resources has mirrored improvements in whole-genome sequencing technologies. These resources enable computationally demanding workflows to be executed in a distributed and on-demand manner, without the up-front costs associated with dedicated HPC resources. Serverless computing is a relatively new paradigm within cloud services, where computation transcends designated resources and follows a pay-for-what-you-use model. It has the potential to reduce computational costs by avoiding the need for dedicated server instances for the duration of workflow execution, development and testing, and can reduce the overhead of cluster management. By providing a simpler method

for utilizing cloud resources, serverless computing has the potential to make on-demand large-scale distributed computing available for a broader category of users, and also promotes reproducibility and portability of workflows.

Our implementation of a best-practice joint variant calling pipeline in SWEEP demonstrates that the serverless paradigm is a promising avenue for executing genomics workflows. For the problem sizes considered, the service remained stable in terms of performance and scaling. However, we also demonstrated that serverless workflows can be limited by the computational ceilings of the CP. These limitations may require adaptation of workflows to minimize disk and memory use of individual tasks, and are a notable disadvantage in comparison with HPC or dedicated cloud resources. For workflows such as joint variant calling, in which there are tasks whose computational requirements grow proportionally with problem size, the capacity of serverless services sets an upper limit to workflow scaling. The increasingly broad services offered by CPs will, however, likely render serverless computing a more versatile paradigm with time.

## Supporting information

**S1 Text.**
(TXT)

## Acknowledgments

We acknowledge the eScience Institute at the University of Washington and the Microsoft AI for Earth program for providing our team with cloud credits to run SWEEP workflows.

## Author Contributions

**Conceptualization:** Aji John, Kathleen Muenzen, Kristiina Ausmees.

**Data curation:** Aji John, Kathleen Muenzen.

**Formal analysis:** Aji John, Kathleen Muenzen, Kristiina Ausmees.

**Investigation:** Aji John, Kathleen Muenzen, Kristiina Ausmees.

**Methodology:** Aji John, Kathleen Muenzen, Kristiina Ausmees.

**Project administration:** Aji John.

**Software:** Aji John, Kathleen Muenzen, Kristiina Ausmees.

**Visualization:** Kristiina Ausmees.

**Writing – original draft:** Aji John, Kathleen Muenzen, Kristiina Ausmees.

**Writing – review & editing:** Aji John, Kathleen Muenzen, Kristiina Ausmees.

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
