## [Decision Letter · Decision Letter 0]

22 Dec 2020

PONE-D-20-35172

Evaluation of serverless computing for scalable execution of a joint variant calling workflow

PLOS ONE

Dear Dr. John,

Thank you for submitting your manuscript to PLOS ONE. After careful consideration, we feel that it has merit but does not fully meet PLOS ONE’s publication criteria as it currently stands. Therefore, we invite you to submit a revised version of the manuscript that addresses the points raised during the review process.

We look forward to receiving your revised manuscript.

Kind regards,

Jacopo Soldani

Academic Editor

PLOS ONE

"The funders had no role in study design, data collection and analysis, decision to publish, or preparation of the manuscript"

"KA and AJ are developers of SWEEP , the workflow tool used in the publication"

Reviewers' comments:

Reviewer's Responses to Questions

**Comments to the Author**

1. Is the manuscript technically sound, and do the data support the conclusions?

Reviewer #1: Partly

Reviewer #2: Yes

2. Has the statistical analysis been performed appropriately and rigorously? 

Reviewer #1: I Don't Know

Reviewer #2: N/A

3. Have the authors made all data underlying the findings in their manuscript fully available?

Reviewer #1: Yes

Reviewer #2: Yes

4. Is the manuscript presented in an intelligible fashion and written in standard English?

Reviewer #1: Yes

Reviewer #2: Yes

5. Review Comments to the Author

Reviewer #1: ## Information on the Contribution

Manuscript ID: PONE-D-20-35172

Title: "Evaluation of serverless computing for scalable execution of a joint variant calling workflow"

Author(s): Aji John, Kathleen Muenzen, Kristiina Ausmees

Summary:

In the presented work, authors describe the modeling and execution of GATK best-practice joint variant calling pipeline using SWEEP, a process engine offered as a service, which supports executing tasks hosted on AWS and Azure. In particular, SWEEP focuses on the serverless-style execution, meaning that it supports invoking business logic hosted using FaaS/CaaS offerings from AWS and Azure. The workflows are defined using a custom DSL which also provides control flow constructs for implementing composite integration patterns such as scatter-gather. The main contribution of this paper is the description of the modeled workflow and results/pitfalls of its execution using SWEEP in combination with AWS/Azure. In general, the paper reads well and provides some interesting insights related to the topic of serverless workflows. However, the publication has inaccuracies related to the description of research design and the novelty of presented insights, which affects the overall quality of the work.

## Comments

1. Authors need to highlight the difference from the previously published work (John et al. "SWEEP: Accelerating Scientific Research Through Scalable Serverless Workflows")

The original work that introduces SWEEP also presents a case study (Section 4.1) which focuses on modeling the variant calling workflow. From the workflow modeling perspective, the workflow presented in this paper is not significantly different from the already published case study. It would be helpful if authors emphasize how the workflow presented in this paper is different. Additionally, since no source code was referenced in the previously-published work, for readers not from the bioinformatics field it might look like a small delta from the previously-published work.

2. Authors need to provide more technical details about the workflow execution

* Both in the paper and in the GitHub repository authors state that the modeled workflow was executed on AWS and Azure. However, in the Discussion section (page 12, line 222) authors mention that due to technical limitations it was not possible to execute some tasks hosted on AWS. It is not quite clear, which tasks were executed on which provider and how this influenced the modeled workflow(s). It would be helpful if authors emphasize which actions are required from users to model-deploy-execute a SWEEP-based workflow, and how AWS/Azure come into play here. In case some tasks from the modeled workflow were executed ONLY on Azure, this means that the actual workflow is not a single-cloud workflow and, among other issues, requires considering costs of inter-cloud data transfers too.

* The implementation requirements related to function/container code are not clear. Since AWS and Azure impose different requirements on source code, integration with service offerings, and packaging formats, is it the case that SWEEP workflows can only be enacted on multiple providers if the code does not use provider-specific features at all? In addition, the invocation of FaaS functions can happen differently on AWS and Azure, e.g., HTTP-based using API Gateways, events, direct calls from the source code. The question is which requirements SWEEP workflows impose on the function code. In case API Gateways are used to invoke functions, the costs of API Gateway offerings have to be included into the picture as well, similar to data transfer costs if cross-cloud interactions were needed for enacting the workflow.

* It is stated that SWEEP workflows require actual functions/containers to be already deployed on the target provider, although its API has endpoints for uploading functions. As stated before, it would be helpful if authors clarify the actions required from users to model/deploy (both workflow and functions/containers), and execute a SWEEP workflow.

3. Authors need to explain why using SWEEP is more beneficial than using provider-specific orchestrators such as AWS Step Functions(SF) and Azure Durable Functions(DF)

Both SF and DF are mature orchestrators supporting using FaaS/CaaS offerings from AWS and Azure which provide out-of-the-box service integration, error handling, etc.

From the description of the process it is not clear, why SWEEP is more beneficial to use.

Firstly, constructs allowing implementing composite integration patterns such as scatter-gather are present in SF and DF, together with means to automate the deployment of code/workflows, whereas this work has to be done manually in SWEEP. Moreover, since it is not quite clear which communication type is used to trigger functions/containers, the costs of using SWEEP might incur not only the per-request costs, but also additional service offerings such as AWS API Gateway, making the overall costs more expensive than using SF, for example. In the best case, authors might provide a comparison between SWEEP and solutions from AWS/Azure to highlight pluses/minuses of using SWEEP workflows, e.g., cross-cloud workflow orchestration becomes possible, bioinformatics pipelines execution is more straightforward, etc.

4. Phrasing:

-- page 3, line 35: "As SWEEP is fully built on the serverless framework, ..."

This is confusing, Serverless framework is a deployment automation tool, whereas SWEEP is presented as a serverless orchestrator / workflow management system.

-- page 4, line 64: "... using the Function-as-a-Service (FaaS) paradigm."

-- page 4, line 66: "... using Container-as-a-Service (CaaS) services."

"cloud service model" instead of paradigm/services?

Reviewer #2: The authors contribute a work in the area of workflows based on cloud functions. This is an industrially relevant area as evidenced by serverless workflows offered commercially by cloud providers.

The domain for applying the workflows is genomics, in particular genome sequencing with joint-variant workflows. It is not the first article discussing the combination of serverless computing and bioinformatics. In 2019, Niu et al. studied high-performance sequence comparison, and Crespo-Cepeda et al. investigated the potential of the combination in general. Unfortunately, neither of these works are referenced so it remains unclear how much overlap and novelty there is.

SWEEP uses functions and containers to execute workflow tasks. It is not clear if CaaS refers to traditional long-running container services (e.g. pure Docker or Kubernetes), or to short-lived executions (e.g. Fargate, Google Cloud Run, Knative), and whether the statelessness is hence an issue for both. Fargate and ACI are mentioned later on, yet it is not clear why all tasks per runtime get the same memory allocation, and why the memory allocated to functions is much smaller (e.g. 3 GB instances would be possible with Lambda and would speed up execution). That should all be clarified in the section on definition of workflows. More technical details on the "as opposed to Docker Hub base images" would also be useful - whether it merely concerns the installation of additional packages, or also the interface to the containers and how information is passed to them and results are retrieved. It looks like in total (Fig. 1) 5 out of 18 workflow steps are cloud functions, whereas much the paper makes it sound as if most of the workflow was. What were the decisions to go for either technology in each step? This knowledge would greatly contribute to the value of the manuscript as it would help readers facing similar challenges.

Fig. 2 positions a cost of zero to a runtime of around 3 hours. First, the graph starts at around 15000s, or around 4 hours; and furthermore, it would imply 0 or negative cost for shorter runtimes whereas the pricing model of FaaS/CaaS is linear except for free tiers. It is not clear if the graph was created and interpreted correctly. Fig. 2 is also not referenced from the text. Some attention should be given to it because it represents the value proposition of the paper, that using SWEEP would be cost-effective.

The manuscript as a whole could be made more readable. The acronym GVCF is not explained; presumably the G stands for Gathering but the rest is unclear. What does it do and why did you choose it as representative workflow that would allow to generalise findings? A typical JSON excerpt of GVCF in your own SWEEP language would further help in understanding the nature of the workflow and how it would map to cloud functions or containers. Moreover, most providers - including AWS - offer their own languages for serverless workflows; a short statement of differentiation would also be helpful (that might focus on portability and ability to also include containerised endpoints).

Nitpicks: Fig. 1 has dashed edges but the caption talks of dotted edges.

Datasets and sample workflows are made available and look reasonable.

For transparency reasons, it would be helpful to point out in the review materials that SWEEP is linked to a startup. KM is listed on the SWEEP website as lead software engineer but the competing interests only mention KA and AJ.

6. PLOS authors have the option to publish the peer review history of their article (what does this mean?). If published, this will include your full peer review and any attached files.

Reviewer #1: No

Reviewer #2: No

---

## [Author Response · Author response to Decision Letter 0]

6 Mar 2021

Thank you for your insightful comments on our submitted manuscript, “Evaluation of serverless computing for scalable execution of a joint variant calling workflow.” We have reviewed the comments and added clarifications to the manuscript as needed. Please see our responses (in bold) to each reviewer comment (in italics) below.

After adding these proposed changes, we believe this paper is now suitable to be published in PLOS One. Thank you again for your consideration.

Sincerely,

Aji John

Kristiina Ausmees

Kathleen Muenzen

---

## [Decision Letter · Decision Letter 1]

6 Apr 2021

PONE-D-20-35172R1

Evaluation of serverless computing for scalable execution of a joint variant calling workflow

PLOS ONE

Dear Dr. John,

Thank you for submitting your manuscript to PLOS ONE. After careful consideration, we feel that it has merit but does not fully meet PLOS ONE’s publication criteria as it currently stands. Therefore, we invite you to submit a revised version of the manuscript that addresses the points raised during the review process.

We look forward to receiving your revised manuscript.

Kind regards,

Jacopo Soldani

Academic Editor

PLOS ONE

Journal Requirements:

Additional Editor Comments (if provided):

Please address the minor comments provided by the reviewers in preparing the final version of your manuscript.

Reviewers' comments:

Reviewer's Responses to Questions

**Comments to the Author**

1. If the authors have adequately addressed your comments raised in a previous round of review and you feel that this manuscript is now acceptable for publication, you may indicate that here to bypass the “Comments to the Author” section, enter your conflict of interest statement in the “Confidential to Editor” section, and submit your "Accept" recommendation.

Reviewer #1: (No Response)

Reviewer #2: All comments have been addressed

2. Is the manuscript technically sound, and do the data support the conclusions?

Reviewer #1: Yes

Reviewer #2: Yes

3. Has the statistical analysis been performed appropriately and rigorously? 

Reviewer #1: N/A

Reviewer #2: N/A

4. Have the authors made all data underlying the findings in their manuscript fully available?

Reviewer #1: Yes

Reviewer #2: Yes

5. Is the manuscript presented in an intelligible fashion and written in standard English?

Reviewer #1: Yes

Reviewer #2: Yes

6. Review Comments to the Author

Reviewer #1: Manuscript ID: PONE-D-20-35172_R1

Title: "Evaluation of serverless computing for scalable execution of a joint variant calling workflow"

Author(s): Aji John, Kathleen Muenzen, Kristiina Ausmees

In this work, the authors presented the modeling and execution of GATK best-practice joint variant calling pipeline using SWEEP, a process engine offered as a service, which supports executing tasks hosted on AWS and Azure.

The revision addresses issues raised by reviewers related to the original submission.

The difference between the previous work and the workflow presented in this work is explained.

Authors also provide clarifications on how cloud provider services were used, highlighting that this workflow can only be executed in a multi-cloud setting due to limitations of certain cloud services.

Authors provide a high-level clarification on interaction with provider-hosted function/containers, which is probably motivated by SWEEP commercialization plans. While the aspects of interaction with provider-specific services are not directly required in the context of this work, such technicalities influence the code implementation requirements. In particular, to achieve the vendor-agnosticism mentioned as one of the main motivations for using SWEEP, functions/containerized applications might have to be developed without relying on provider-specific libraries. For example, if an Azure-specific Java annotations are used in a function, deploying it to AWS Lambda would require additional efforts (or does the system handle such cases for users?). The documentation also does not specify how functions/containerized applications need to be implemented, e.g., implementing certain interfaces, adhering to certain data formats. In my opinion, highlighting such requirements would strengthen the work.

Overall, I am glad to accept this work when more details on code implementation requirements are provided.

Reviewer #2: This is a revised manuscript that addresses most of the previously raised concerns. Indeed, the motivation for using SWEEP given the characteristics of the bioinformatics/genomics workflow needs is now more clear, and the authors contribute a critical view on using serverless platforms for the workflow steps not requiring these execution features. The nature of the work has not changed, it remains an experience report discussing and experimentally measuring the SWEEP migration of a given pipeline. Such experiences help the serverless systems community to achieve more applicable system designs. I have no major points to complain, but if another round of minor edits is possible, there are some more rough edges that could improve with more attention.

A critical point is the limitation on number of tasks. If each task stays within the specified disk limit and yet the task fails (and requires retries), that might be labelled more explicitly a bug in ACI, or it is some other effect that should be explained in greater detail. Given the potential blocker impact on running other large workflows, it would also be interesting to know if for instance Fargate suffers from the same effect. The preference for ACI over Fargate is explained with the greater disk size (50 vs 10 GB), but that greater disk may not be worth much when the instances fail randomly.

Adaptive selection of storage is mentioned as possible extension of the work. It would be great if some hints were given how this could be done in a way that the workflow will benefit, for instance by accomodating some latency issues by juggling between instance disk, externally mounted disk (on AWS Lambda) and managed storage (AWS S3/ABS) equivalent.

A useful addition, if space permits, would be an elaboration on the modifications of container images compared to those found on Docker Hub. Specifically, if this merely concerns the ability to invoke them in as ACI, or any other changes beyond the invocation interface.

7. PLOS authors have the option to publish the peer review history of their article (what does this mean?). If published, this will include your full peer review and any attached files.

Reviewer #1: No

Reviewer #2: No

---

## [Author Response · Author response to Decision Letter 1]

20 May 2021

Dear Reviewers and Editors,

Thank you for your insightful comments on our submitted manuscript, “Evaluation of serverless computing for scalable execution of a joint variant calling workflow.” We have reviewed the recent comments and added clarifications to the manuscript and to our documentation as needed. Please see our responses (in bold) to each reviewer comment (in italics) below.

After adding these proposed changes, we believe this paper is now suitable to be published in PLOS One. Thank you again for your consideration.

Sincerely,

Aji John

Kristiina Ausmees

Kathleen Muenzen

Reviewer 1 Comments:

In this work, the authors presented the modeling and execution of GATK best-practice joint variant calling pipeline using SWEEP, a process engine offered as a service, which supports executing tasks hosted on AWS and Azure.

The revision addresses issues raised by reviewers related to the original submission.

The difference between the previous work and the workflow presented in this work is explained.

Authors also provide clarifications on how cloud provider services were used, highlighting that this workflow can only be executed in a multi-cloud setting due to limitations of certain cloud services.

Authors provide a high-level clarification on interaction with provider-hosted function/containers, which is probably motivated by SWEEP commercialization plans. While the aspects of interaction with provider-specific services are not directly required in the context of this work, such technicalities influence the code implementation requirements. In particular, to achieve the vendor-agnosticism mentioned as one of the main motivations for using SWEEP, functions/containerized applications might have to be developed without relying on provider-specific libraries. For example, if an Azure-specific Java annotations are used in a function, deploying it to AWS Lambda would require additional efforts (or does the system handle such cases for users?). The documentation also does not specify how functions/containerized applications need to be implemented, e.g., implementing certain interfaces, adhering to certain data formats. In my opinion, highlighting such requirements would strengthen the work.

Overall, I am glad to accept this work when more details on code implementation requirements are provided.

Our response

---------------

The discussion regarding provider-specific libraries and how this affects the use of SWEEP is an interesting one, but one that we feel is out of scope for this paper. The main goal of the study at hand is to discuss the use of SWEEP to implement this particular workflow, and not a description of the framework itself. We have therefore tried to keep details about SWEEP to the minimum that is required for understanding the implementation and focused on the variant calling application. However, this comment is very pertinent to aspirations of any workflow provider, and certainly does apply to SWEEP. We have since incorporated some of the suggestions in our documentation on SWEEP website regarding implementation details for function/container tasks, and plan to continuously improve it by providing more examples. 

Reviewer 2 Comments:

This is a revised manuscript that addresses most of the previously raised concerns. Indeed, the motivation for using SWEEP given the characteristics of the bioinformatics/genomics workflow needs is now more clear, and the authors contribute a critical view on using serverless platforms for the workflow steps not requiring these execution features. The nature of the work has not changed, it remains an experience report discussing and experimentally measuring the SWEEP migration of a given pipeline. Such experiences help the serverless systems community to achieve more applicable system designs. I have no major points to complain, but if another round of minor edits is possible, there are some more rough edges that could improve with more attention.

A critical point is the limitation on number of tasks. If each task stays within the specified disk limit and yet the task fails (and requires retries), that might be labelled more explicitly a bug in ACI, or it is some other effect that should be explained in greater detail. Given the potential blocker impact on running other large workflows, it would also be interesting to know if for instance Fargate suffers from the same effect. The preference for ACI over Fargate is explained with the greater disk size (50 vs 10 GB), but that greater disk may not be worth much when the instances fail randomly.

Adaptive selection of storage is mentioned as possible extension of the work. It would be great if some hints were given how this could be done in a way that the workflow will benefit, for instance by accomodating some latency issues by juggling between instance disk, externally mounted disk (on AWS Lambda) and managed storage (AWS S3/ABS) equivalent.

A useful addition, if space permits, would be an elaboration on the modifications of container images compared to those found on Docker Hub. Specifically, if this merely concerns the ability to invoke them in as ACI, or any other changes beyond the invocation interface.

Our response

---------------

Regarding the issues if failing tasks and the causes, a paragraph was added in order to discuss the failure of tasks in ACI. While it would be interesting to investigate the causes of the failures in more detail in the future, this is considered out of scope for the current paper. We instead aim to describe it from a user perspective, as an unexpected issue that we came across when using the services of a particular CP. Also, while previous experiments suggest that Fargate is more stable w.r.t this particular issue, we do not have data supporting this for this particular workflow and tasks, and therefore refrain from a direct comparison. We instead added a more general discussion on the stability of serverless services that a user might find relevant to consider. Please see lines 233-238 discussing these points.

For comment w.r.t to adaptive selection of storage, we note that filesystem mounting capability has since become available for functions and container-based tasks; this might be a good option to explore when disk size available for compute is limited. Lines 254-257 were added to elucidate this point.

We have also added details on SWEEP website on porting of Docker images.

---

## [Decision Letter · Decision Letter 2]

25 Jun 2021

Evaluation of serverless computing for scalable execution of a joint variant calling workflow

PONE-D-20-35172R2

Dear Dr. John,

We’re pleased to inform you that your manuscript has been judged scientifically suitable for publication and will be formally accepted for publication once it meets all outstanding technical requirements.

Kind regards,

Jacopo Soldani

Academic Editor

PLOS ONE

Additional Editor Comments (optional):

Reviewers' comments:

Reviewer's Responses to Questions

**Comments to the Author**

1. If the authors have adequately addressed your comments raised in a previous round of review and you feel that this manuscript is now acceptable for publication, you may indicate that here to bypass the “Comments to the Author” section, enter your conflict of interest statement in the “Confidential to Editor” section, and submit your "Accept" recommendation.

Reviewer #1: All comments have been addressed

Reviewer #2: All comments have been addressed

2. Is the manuscript technically sound, and do the data support the conclusions?

Reviewer #1: Yes

Reviewer #2: Yes

3. Has the statistical analysis been performed appropriately and rigorously? 

Reviewer #1: N/A

Reviewer #2: N/A

4. Have the authors made all data underlying the findings in their manuscript fully available?

Reviewer #1: Yes

Reviewer #2: Yes

5. Is the manuscript presented in an intelligible fashion and written in standard English?

Reviewer #1: Yes

Reviewer #2: Yes

6. Review Comments to the Author

Reviewer #1: (No Response)

Reviewer #2: This work demonstrates experimentally and analytically how to run a genome sequencing workflow with a new workflow engine called SWEEP. The contribution is mostly in the analytical nature, because the workflow engine itself is not new. The article can be read easily, the covered technologies are relevant subjects for investigation, and the use case of genome sequencing is of societal importance.

According to the response letter and highlighted changes, the authors have addressed a few issues that were left from previous iterations. It looks like the modifications are almost too brief or superficial (on pages 10 and 11), but given that no major changes were requested from them, the present manuscript is fine for me to be published.

7. PLOS authors have the option to publish the peer review history of their article (what does this mean?). If published, this will include your full peer review and any attached files.

Reviewer #1: No

Reviewer #2: No

---

## [Editor Report · Acceptance letter]

29 Jun 2021

PONE-D-20-35172R2 

Evaluation of serverless computing for scalable execution of ajoint variant calling workflow 

Dear Dr. John:

I'm pleased to inform you that your manuscript has been deemed suitable for publication in PLOS ONE. Congratulations! Your manuscript is now with our production department. 

Kind regards, 

on behalf of

Dr. Jacopo Soldani 

Academic Editor

PLOS ONE